# Birds of Prey, Birds of Wisdom: Relating to Non-Humans in Contemporary Western-Based Shamanism

Carolina Ivanescu *  and Nienke Groskamp

Department of Religious Studies, University of Amsterdam, Kloveniersburgwal 48,
1012 CX Amsterdam, The Netherlands
*  Correspondence: c.ivanescu@uva.nl

**Abstract:** Birds of prey appear frequently in contemporary forms of shamanism. For example, Michael Harner's Core Shamanism references the 'power animal,' or the authentic self, which sometimes takes the form of a strong and benevolent eagle. However, precisely how meaning and belief concerning these birds may have been lost, challenged or (re)invented remains to be explored. In this contribution, we have used the methods and vision of netnography to explore the relationships between contemporary western-based, self-defined shamans and birds of prey: real, imagined or represented.

**Keywords:** neoshamanism; practice of shamanism; tradition and innovation; construction of meaning; netnography

## 1. Introduction

In a text considered foundational to the study of shamanism, Mircea Eliade recounts the following Buryat legend:

> The gods decided to give mankind a shaman to combat disease and death, and they sent the eagle. But men did not understand its language; besides, they had no confidence in a mere bird. The eagle returned to the gods and asked them to give him the gift of speech, or else to send a Buryat shaman to men. The gods sent him back with an order to grant the gift of shamanizing to the first person he should meet on earth. Returned to earth, the eagle saw a woman asleep under a tree, and had intercourse with her. (Eliade [1964] 1972, p. 69)

Before and since Eliade's account, the importance of birds, especially birds of prey in 'traditional,' culturally specific shamanic practices, has been well-documented (See Walter and Fridman 2004). Both in Siberia and South America, for example, bird spirits have long accompanied shamans as they traverse invisible worlds. In this context, the bird has been described as a guardian, a guide, a helper, or the form the shaman assumes to enter an altered state, by practitioners such as Christina Pratt (Pratt 2007). Birds of prey also play a central role in manifold shamanic cosmologies, usually as spirits from the Upper World (Vinogradov in Walter and Fridman 2004).

They are also present in shamanic initiations across the globe, for instance among the Unmatjera tribe and the Warramunga, who tell of these birds dismembering the bodies of neophytes and then assisting in their successive reconstitution (Eliade [1964] 1972). The birds additionally serve as guides or psychopomps in liminal states of being, or in 'journeys' between different realms, which are characteristic of many forms of shamanism. Mirroring the traits of these birds physically, metaphorically and/or spiritually, shamans describe experiences in the spirit world of floating, flying and travelling out-of-body (Bahr in Walter and Fridman 2004). The Akawaio of Colombia, for example, believe that songs are the wings of the shaman's magical flight; various bird spirits lend the shaman their 'wings' in the form of specific songs during trance (Shepard Jr. in Walter and Fridman 2004).

Furthermore, becoming a bird or being accompanied by one may indicate the capacity, while still alive, to undertake an ecstatic journey into the sky and the beyond (Eliade [1964] 1972).

In the examples above, we see that the connection with and knowledge of birds of prey can be culturally specific in many instances. However, this broader relationship has also been identified as universal and trans-cultural, especially in the linkage of birds of prey to the spirit world (Eliade [1964] 1972). Present-day forms of shamanism often draw from seemingly trans-cultural tropes, appearing cosmopolitan and universalistic in tone (Vitebsky 1995) while nonetheless eclectically incorporating traditional shamanic elements (Atkinson 1992; Blain and Wallis 2000; Lindquist 1997). At times, they claim 'a revival of ancient wisdom' (Vitebsky 1995, p. 196). In our contemporary society, people "who consider themselves to be shamans or to be doing 'shamanic' things" (Harvey 1997, cited in Wallis 2003) are a rather large and diverse group. These practices arise in urban environments, but as mentioned before, they are at times connected to specific cultures and in some cases build on a mixture of varied traditions. They are known to the research literature as neo-shamanism (Wallis 1999; Davidov 2010), of which Core Shamanism is a subcategory, presenting itself as a simplified and universalistic practice (Johnson 1995; Townsend 2004). Both are forms of 'modern shamanic spirituality' (Townsend in Walter and Fridman 2004, pp. 49–58), and we will use this term throughout the article. The processes by which modern forms of shamanism draw upon (a combination of) tradition(s) lead us to the expectation that birds of prey remain important in the contemporary context as well. But what are the possible ways that present-day shamans in the west relate to and with birds of prey? This is the core question we will engage with.

In our research, we refer to traditional shamanism as an emic category that informs the self-identification and practices of neo-shamans. As a scholarly concept, shamanism has been criticized due to its constructed or 'invented' nature, but nonetheless the tendency to universalize shamanism as one of the oldest forms of religion is still very much present in contemporary practices (Noel 1997; Wallis 2003; Znamenski 2007; Boekhoven 2013). The problems, which are connected to the definition of shamanism and the influence of different definitions on its study and practice, are enormous and beyond the scope of this article. Suffice it to say, the idea of shamanism as an archaic technique comes from Mircea Eliade (Eliade [1964] 1972) and has since been criticized in scholarly circles. However, for contemporary shamans who concern themselves with traditions for the purpose of legitimizing practices and building community, Eliade's universalizing legacy remains crucial, as well as Harner's emphasis on specific and universal shamanic techniques. Shamanic traditions are, from neo-shamanic points of reference, being 'revived, rediscovered or reinvented' (Blain and Wallis 2000, p. 395) as part of one's personal 'identity business' (Comaroff and Comaroff 2009, p. 74) and as inspiration for new religions of the self (Heelas 1996).

Birds are the anthropological helper spirits of this article; they are mediators guiding us into conceptual spaces that cannot and need not be homogenized (Balzer 1996). Through the thematic lens of birds of prey, we reflect on emic and etic categories, kinship, agency and reciprocity. We thus explore both forms of relating between human and non-human entities and the intentionality behind this relating, keeping a close eye on how agency is expressed, attributed or explained. In this discussion, we will consider agency as the power to act intentionally and ritually, or to have 'spirit charge,' as Lavrillier proposes in her analysis of the relationships between animals and humans among Siberia's Evenki population (Lavrillier 2012). We also take heed of Callon and Law's suggestion that the two conditions of agency in contemporary western culture are 'intentionality and language use' (Callon and Law 1997, p. 491). In our research we take note of language in the larger context of communication. This brings us to the even more profound question of what it means to be human or non-human in today's shamanic practices. We will gently address this question throughout.

## 2. Materials and Methods

*Contemporary Shamans and Birds of Prey*

Birds of prey appear in many contemporary forms of shamanism. For example, Michael Harner's Core Shamanism references the 'power animal,' or authentic self, which sometimes takes the form of a strong and benevolent eagle (Harvey and Wallis 2007). Furthermore, the Eagle Feather Award, which recognizes 'significant contributions to the field of shamanism,' is bestowed by the well-known Society for Shamanic Practice (Society for Shamanic Practice 2022). However, precisely how, and in which ways, meaning and belief concerning these birds may have been lost, challenged or (re)invented remains to be explored. In this contribution, we use the methods and vision of netnography (Kozinets 2019) to explore the relationships between contemporary western-based, self-defined shamans and birds of prey: real, imagined or represented.

This work is based on an online survey consisting of 21 questions, of which 17 were open-ended (see Appendix A). The other four were multiple choice and referred to demographic details of the respondents. We posted this survey in eight moderated, closed Facebook groups concerning shamanism and its practice. Participation was completely voluntary, including the extent to which respondents completed the sequence of questions. All 48 respondents self-defined as shamans or shamanic practitioners, and they filled out the survey in the short interval between the 28 December 2020 and the 6 January 2021. For simplicity, we refer to our respondents as shamans throughout this article, though we touch upon the nuances of identifying as a shaman versus a shamanic practitioner. Their answers have been assessed following the method of content analysis (Badzinski et al. 2021). We took great care to formulate our questions in an open-ended manner and to practice open coding. It should be noted, however, that we were interested in the adherence to and performance of shamanic worldviews, references to culturally specific shamanic traditions, and relationships to birds of prey in a shamanic context. These interests therefore guided our questions and our subsequent interpretations of the answers.

## 3. Results

The first two and final seven questions of the survey focused on shamanic affiliation, shamanic education and demographic information such as age, gender and nationality. Respondents were free to skip questions throughout. In terms of demographics, the overwhelming majority of respondents lived in the United States (24), followed by the United Kingdom (13) and Canada (5). Sweden, Japan, Romania, Australia and The Netherlands were each mentioned once as a country of residence. Almost all respondents were born in North America or the United Kingdom, and many shared being of European descent, with some exceptions including Japanese (1), mixed race (1) and Native American (1). Notably, some respondents answered the demographic question in terms of spiritual belonging: 'the earth is my home,' one wrote, while another wrote that they 'resonate deeply with many First Peoples,' despite the fact that they were born in the United States and were of Western European descent.

We can confidently say that our pool consisted of western-based shamans. Because scholarly literature on neo-shamanism often mentions western and urban shamanism together, we wanted to know whether our respondents lived mostly in urban environments. The results here were varied: 14 lived in a city, 15 in a smaller town and 15 in the countryside. Three respondents selected 'other.' Most could be considered middle-aged: the largest group was between 55–64 years old (17), while the neighboring categories 45–54 and 65–74 were each selected eight times. Nine respondents were between 35–44, and none were older than 75 or younger than 18. In terms of gender, 27 chose 'female,' 18 chose 'male' and three chose 'other.' Most respondents claimed a bachelor's degree (13) as their highest level of education, followed by 'some college credit, no degree' (8) and a master's degree (7).

We were also interested in the shamanic identification of the survey respondents. Interestingly, only 12 were comfortable describing themselves as a shaman; 29 preferred 'shamanic practitioner' instead. Another 12 selected 'other,' and their explanations included

'shaman to be' (R 45 and R 50), 'just a person with some gifts trying to put them to good use' (R 44) and 'Shamanic Christian' (R 1). Most had received formal education as a shaman (21), but many had not (18) and 10 chose 'other'. Their explanations included initiation in a local group (R 10), workshops and courses (R 27) and one-on-one teaching (R 44).

As an entry point into our research, we let respondents free-associate on the topic of birds of prey. After two initial multiple-choice questions about their shamanic identification and affiliation, we asked them what comes to mind when birds of prey are mentioned. Most respondents named specific birds, at times describing them in detail. These details were sometimes zoologic, sometimes spiritual or metaphorical, and often both. Many respondents mentioned particular characteristics of birds, such as their majestic character, their superior eyesight and their impressive strength. At least one respondent made an explicit connection between the physical and material characteristics and the birds' spiritual powers, noting that 'when they look at you it looks like they are looking into your very soul' (R 36). Similarly, another respondent linked the birds' physical and spiritual powers by claiming they fly 'close to higher intelligence' (R 17).

Other respondents singularly and explicitly referred to the supernatural characteristics of birds of prey, seeing them as being either (the same as) spirit(s) (R 22), a sign of spirit (R 32) or 'medicine' (R 21). Still others referred to specific personal circumstances in which birds of prey have featured prominently in their shamanic world as 'healing spirits' (R 4), as escorts to the Upper World (R 6), as solicitors in dreams seeking collaboration (R 15) or as 'animal guides' (R 23). One respondent mentioned actually working with physical, glove-trained birds of prey, implying a relationship of care and domestication, as the birds were described as 'non-releasable' (R 36).

When asked about which species the respondents associate with the term 'bird of prey,' they most often cited eagles, hawks and owls (respectively, 35, 34 and 25 mentions among 50 replies). Falcons, vultures, ospreys, buzzards and condors were also mentioned multiple times (10, 6, 5, 4 and 3 times, respectively), and a few respondents mentioned crows, ravens and kites (2, 3 and 3 times, respectively). In their responses to the free-associative questions, we observe two notable ways in which they taxonomized these birds: first and foremost, based on zoological and quasi-scientific qualities, and second, more sparingly, based on 'archetypal' criteria described by one respondent (R 33) as an unconventional or 'spiritual' way of thinking. This latter basis was also exemplified by another respondent who mentioned the deity Garuda as a bird of prey (R 46).

These two paradigms, zoological and archetypal, shift and blend throughout the responses, existing in a constant tension. To understand the context of these, we asked respondents to describe the importance of (specific) birds of prey within the shamanic tradition(s) they felt connected to. Most respondents expounded in detail on specific birds of prey, stating, for example, that the 'condor and eagle represent winds of the east' (R 41), that the 'eagle is strength,' that the 'hawk is life sustaining own knowledge' (R 28), and that 'the owl allows you to see all sides of a situation at once' (R 48). Others spoke about the significance of birds of prey in general, commenting, for instance, that 'every bird has significance' because 'birds of prey are seen as being closer to the sky spirits' (R 32). Four respondents, however, were both specific and personal in their answers. One stated that the 'golden eagle assists me in psychopomp work' (R 36), while another wrote, 'I am often visited by eagles, in this plane and the others that I journey to' (R 25). These responses combine specific knowledge with personal experience and impression. The descriptions resemble Lavrillier's concept of 'spirit charge,' or the ritual power attributed to each type of non-human (Lavrillier 2012). In our case, the differentiation between them is based on specific knowledge or a combination of specific and personal knowledge. Taking this a step further, we observe 'the attribution of intentionality to certain animals' (Lavrillier 2012, p. 113) but also the importance of the 'personal,' which can be understood as part of the process of individuation of religion in modern times and as an emphasis on one's agency in creating one's own knowledge through personal bricolage.

But which traditions did the respondents feel most connected to? The results here were varied. The largest singular group consisted of four respondents, who identified themselves as belonging to Core Shamanism, referring to Michael Harner's teachings. However, along people who used direct reference to Core Shamanism to identify themselves, we found direct references to teachings or use of core shamanic terms in the answers of another ten respondents. Indeed, one mentioned Harner by name (R 3). Others subscribed to different 'culturally specific' forms of shamanism such as Celtic (R 49, 27, 23), Norse (R 37), Tibetan (R 46) and Andean (R 5). Among this group, 15 respondents practiced, followed or mixed two or more traditions, for example Tibetan Buddhist, Mongolian and Nepalese (R 33). A smaller group (3) mentioned mixing elements from different traditions in 'very non-formal' (R 15) or 'trans-cultural' (R 10) ways. Besides these traditional practices, respondents mentioned being 'taught and instructed by my ancestors' (R 45), being led by 'teachers in the Spirit worlds that search me out' (R 48), being inspired by ancestral 'roots' (R 23), receiving visions while working as a medium (R 28), being guided by 'the energies of the earth' (R 26) and being 'driven' by the land itself (R 34). And birds of prey are seen as connected to the land: 'The land speaks loudly here, and the birds, all of them, not just birds of prey, are her voice' (R 49).

The early questions prompted answers about birds of prey as abstract concepts. To get a better sense of the meaning ascribed to these birds, we also asked whether a sighting could be considered a sign or omen. We deliberately left the interpretation open and left it up to the participants to differentiate between sign and omen, which some did and others did not. The answers showed the same balancing of paradigms, with respondents weighing the symbolic and archetypal meaning with the real-life context of the sighting. According to most respondents (31), seeing birds of prey is indeed an omen or message: 'they represent "heads up: incoming messages" to me' (R 29), especially 'if the bird appeared out of context' (R 26). The message, however, is 'different for each bird' (R 23). Birds are similarly described as signs (R 14, 33), for example, 'a sign to look out as something important for the future is about to happen' (R 47). This can make the sighting feel somehow special: 'If it something [sic] I don't often see but shows up during a certain time or when a certain event is happening, I might see it as a sign' (R 36). The bird's behavior is particularly important here: 'I would only really think about the message if the bird appeared out of context' (R 26). It also depends on 'how it acts' (R 45, 32), for example if it is 'landing: as reminder to ground' (R 8), and which specific bird is concerned (R 32). However, the line between a mundane animal sighting and a bird acting as a messenger/sign is not entirely straightforward. To this effect, one respondent pointed out that 'sometimes they are just natural creatures doing their own thing' (R 27). Thus, the paradigm of interpretation through which the incident is filtered (zoological versus archetypal) is highly context-dependent and individual.

For some, birds of prey act as reminders 'of the essential nature of predator and prey' (R 34), of a personal connection to 'spirit' (R 16) and of a feeling that a 'great spirit is with me and I am on the path' (R 15). However, a sighting's meaning can at times be difficult to parse out. One respondent wrote on bird sightings, 'A sign but don't know how to interpret' (R 13). This problem is sometimes solved pragmatically: 'I like to look them up online and see what jumps out to me to get the message' (R 23). Other times, it can be connected to personal circumstances, especially if one sees 'many in a short period of time' (R 39) or 'during a certain time or when a certain event is happening' (R 36). In this vein, one respondent said that 'if I had just made an offering to the spirits and one flew down low over me, I would take that as a possible omen/sign' (R 33). Another lens for interpreting signs is 'individually in communication with the spirit of the bird' (R 21). While most interpretations are strictly personal, others are also constructed in dialogue with traditions:

> I do [consider a bird of prey an omen], but it is personal for me. For instance, one cultural teaching around hawks is they are a sign of potential intrusion from unwanted spirits in one's life, but I don't take that interpretation unless the hawk I see is doing something to me that communicates this. Otherwise, a hawk up in the sky, or on fence post, is simply just doing what it normally does (R 4).

Here, we notice an emphasis on the individualism of interpretation and meaning-finding, which emphasizes the agency of the shamans themselves in the process of relating to birds of prey.

As symbols, birds of prey function as links between human, natural and spiritual realms, as seen in the examples above (Boykova and Walter in Walter and Fridman 2004). In this way, the semiotic distinction between signified and signifier becomes obsolete for the shamanic practitioner. Perhaps the bird not only carries but embodies or becomes its own spiritual meaning, which is then 'given' to the shaman. This transformation can be considered a postmodern, performative answer to the need for physical transformation, which in scholarly literature has been assumed as a condition for shamanism. Turner's work on the Bororo of Brazil for example, narrates how 'spirit actors' don costumes modeled on, representative of or incorporating aspects of animals or birds. According to Turner, 'the donning of the feather as the ritual costume . . . implies the metonymic acquisition by the dancers of the arara powers ("spirit") metonymically and metaphorically embodied by the feathers,' which are joined by the 'human powers of social and cultural creation and reproduction, metaphorically transformed through the ritual performance' (Turner 1991, p. 147). Since we know that scholarly work on shamanism as a cross-cultural category has been extremely influential to neo-shamanism, it makes sense to think that neo-shamans would be inspired by the idea of transformation and would adapt it to fit their circumstances and personal beliefs. In this case, the relationship between the shaman and the birds is more direct, implying a transfer that is mysterious both in content (what is being transferred) and modality (how it is being transferred). To further assess this, we will look at different modes of transfer between bird(s) and shamanic practitioners, as reflected in the responses we gathered.

**4. Modalities of Transfer**

In our survey responses, we can identify roughly three types of transfer between birds of prey and shamanic practitioners. We refer to these as *transfer through emulation*, *transfer through encounter* and *transfer through physical touch.* Our attempt to map these types of transfer is in conversation with Ojamaa's three types of zoomorphic and ornitomorphic transformations in Siberian shamanism (Ojamaa 1997). According to Ojamaa, Siberian shamans partook in objective transformations (the transformation of objects like drums and costumes into animals), soundic transformation (the transformation of a shaman through imitation of animal sounds) and expressive transformation (the transformation of a shaman through imitation of animal behavior).

Our exploration of transfer is inspired by this model. It is important to note, however, that we are not proposing that our respondents precisely adhered to the theoretical model, which was developed for Siberian shamanism in particular. Rather, we are instead interested in whether they experienced similar transformative encounters with animals, although admittedly our data spoke more to transfer than to transformation. Thus, we also develop our own model of shamanic transfer outlined below. In doing so, we attempt to move away from considering traditional, culturally specific forms of shamanism as the norm, avoiding the assumption that transformation of the self is the primary way our respondents acquire animal powers. Instead, we ask: how do contemporary shamanic practitioners relate to (the powers of) birds of prey?

*4.1. Transfer through Emulation*

A primary type of transfer we see in our survey responses is transfer through emulation. By this, we mean that the shamanic practitioner actively cultivates, takes agency in embodying actively the zoological or archetypal qualities they witness in birds of prey. Exemplifying this, one respondent wrote that '[birds of prey] have inspired me. They helped me get in touch with qualities that I wanted to make conscious and develop in myself' (R 4). Another shared an anecdote: 'One day I was so unsure of what I should do, I needed guidance and all of a sudden I saw a hawk dive and grab a mouse. I took it as a message

to go after what I want. I had never seen a hawk catch prey before' (R 23). This is a clear example of transfer through emulation, as the hawk's action was taken as an indication to press forward with determination. Similarly, some respondents particularly revered birds of prey for their superior eyesight, which one respondent described as 'reminding me to use my vision and strength' (R 8) and another as 'reminding me to take a broader view of a particular situation' (R 47). Interestingly, most responses fitting our transfer through emulation model use words like 'message' and 'reminder' in describing the relationships between birds and the shamanic practitioners.

Notably, the bird does not convey its qualities directly to the shamanic practitioner; rather, it serves as inspiration for the shaman to get in touch with their own inner qualities. We can connect this to Heelas' definition of New Age as according ultimate authority to the self (Heelas 1996). This in turn places the shaman in a position of power and control over the process initiated by the bird. Emulation for the shaman is a choice, as they internalize something initially perceived as outside of the self. According to Townsend (2004), this emphasis on inner qualities over external authority is applicable to neo-shamanism. "For traditional shamanism and Core Shamanism, knowledge and direction come from spirits. For the others, knowledge and direction come from 'within,' from one's higher self, inner voice, or inner wisdom" (Townsend 2004). Despite this claim, transfer through emulation has also been ascribed to indigenous American shamanic practices, including but not limited to the Desana people of the Colombian Amazon, where 'the shaman is able to identify and establish a personal kinship bond with a particular Animal Being, perhaps replicating its character' (Frey in Walter and Fridman 2004). In this case, the shaman becomes an apprentice to the animal, learning to become the animal itself and gradually transforming more or less at will. From both this perspective and that of our survey responses, the most defining characteristic of this type of transfer is its relational character; the bird reaches out to the shaman, who then interprets its message and extracts a meaning, transforming (or not) into the bird itself.

*4.2. Transfer through Encounter*

The bird's agency is even more pronounced in transfer through encounter. This type—which, as per its namesake, takes place through encounters—can be subdivided into roughly two categories: encounters in alternate reality and encounters in physical reality. It should be noted, however, that the boundaries between these categories are analytical and, as such, sometimes overlap.

Encounters with birds in alternate reality often take place when the shaman is 'journeying' or in trance. In this case, birds can function as psychopomps (R 36), ferrying the respondent to the upper world (R 6) or acting as an animal guide (R 23). Sometimes the shaman actively seeks out the birds of prey: 'the spirit world ones help me when I come to them' (R 7), but more often the bird's agency is emphasized: 'I am often visited by eagles, in this plane and the others that I journey to' (R 25). We might note the use of passive voice here, which emphasizes that the bird is in control over the encounter. In these journeys, the practitioner themself can also transform into a bird of prey: 'I often journey in owl form' (R 37), says one respondent, whereas another tells a story in which they 'became an eagle and flew around all the area where I was meditating' (R 9). The latter is a good example of the overlap between alternate and physical realities, echoing Ojamaa's expressive transformation modality (Ojamaa 1997).

The most significant examples of these encounters involve physical birds of prey, which according to our respondents represent or embody spirits. One respondent writes of 'healing spirits that come as birds of prey' (R 4), while another views these birds as 'messengers from spirit and friends in animal form' (R 23). An encounter with a bird of prey can also be seen as confirmation of an earlier spiritual journey: 'Snowy Owl came to me in Journey, I was able to transform and fly as Owl. Shortly after A Snowy Owl came and perched in a tree next to me twice' (R 48). In this case, the bird in alternate reality and the physical bird are seen as connected.

Other times, it is nearly impossible for us to categorize the encounter. The same respondent, for example, also wrote that 'I am able to see through the eyes of the Hawk, allowing me to see things in great detail' (R 48). These are clear examples of transfer, as the practitioner is able to borrow the birds' powers to advance a shamanic journey, but we know little about the circumstances.

These examples testify of social connections between humans and non-humans, connections that are cultivated by both parties, though in different manners. Birds of prey allow themselves to be seen, give gifts, heal, convey messages and allow access to knowledge. In terms of encounters, birds seem to wield more power over humans than the other way around, and the animals determine the extent and content of these connections.

Emulation and encounter exist in close relationship with each other and are regarded as forms of sacred and intimate bonds, which remain imbedded in the body's memory. As contemporary shaman Keeney explains:

> For example, when I had a vision of an eagle facing me on a cliff during the midst of a fasting ceremony, it was such an emotional impact that it became expressed or localized in my arms and shoulders. Now, years later, if I pray and become mindful of that eagle, I may feel that part of my body begin to move, imitating the movement of a bird's wings. I do not purposefully choreograph this movement. It happens without conscious intention. Here the experience is a sacrament of relational connection. (Kottler et al. 2004, p. 52)

### 4.3. Transfer through Physical Touch

Another powerful form of transfer occurs via physical touch and is often connected to owning bird-related items. Most respondents (42) were involved in this type of transfer, mentioning among their items feathers in particular. This echoes the emphasis on feathers that has been attributed to various culturally specific shamanic traditions. In her *An Encyclopedia of Shamanism*, practitioner Christina Pratt narrates several times how feathers feature prominently on crowns and other power objects worn or used ritualistically by shamans. In South America, for example, she claims that feathers are directly connected to birds, which are considered sacred beings and which in turn connect the shaman to the transformative powers of the Divine Sun (Pratt 2007). Wearing feathers facilitates the shaman's shift from human to spirit in trance, she writes, just as animal masks or skins facilitate shapeshifting into more powerful spirit forms (Pratt 2007). As stated above, these accounts of 'traditional' shamanism have their bearing on the practices of neo-shamans. For our purposes, it is less interesting whether this emphasis on feathers in shamanic traditions is justified than it is important to acknowledge the (mythical) importance attributed to feathers by western shamans such as Pratt. This serves as important background against which we interpreted our survey responses.

One respondent mentioned using feathers as a fan for smudging. Other body parts were also claimed as items, including wings, claws, talons, feet, tails and skulls. Some respondents similarly mentioned possessing representations of birds in the form of prints, figurines, statues and books. On the topic of possessions, one respondent even curiously mentioned living on a street named after a bird (R 15). Birds of prey also appear on altars (R 32, 31) in manifold forms. Their physical presence is a reminder of, or a direct connection to, spirit. For example, an eagle statue is connected to the 'helping spirit . . . that helps me in crossing spirits over' (R 36). One of the respondents also described finding a dead great horned owl in the road: 'As I touched it, I could feel owl medicine transfer into me' (R 14). This anecdote alludes to a transfer of 'vital force' as described by Ingold (2000). In this case, the transfer explicitly took place through physical touch. From this and the other examples, we can see that objects and animals are perceived to be alive (or to be a 'being') because people engage in and maintain a relationship with them (Ingold 2000; Bird-David 1999).

Birds of prey and their feathers can sometimes even be found on the practitioner's body itself in the form of a tattoo (R 12). This is a sort of long term, quasi-embodiment, or a part of shamanic ritual armor (R 44, 33), which is reminiscent of the transformative

powers of costumes in traditional shamanism. In Ainu shamanism, for example, wearing an animal's skin, such as a bear's, is interpreted by the shaman as equivalent to acquiring the characteristics and powers of the animal itself (Tanaka in Walter and Fridman 2004). Some ethnic groups in California have also been described as having medicine men called 'bear-doctors' who were believed to transform themselves into bears, their tutelary spirit, by donning the bear's skin (Hultkrantz 1963 cited by Riboli in Walter and Fridman 2004). Our respondents' experiences of transfer through representations, symbols or body parts seem to match these examples described in scholarly literature. They also fit Ojamaa's category of objective transformation (Ojamaa 1997). The material culture of traditional shamanism, including feathers and bones, is replicated, though the meaning and use of these artifacts are highly personalized.

Many respondents explicitly insisted that the feathers and bones in their possession were legal or were merely found, showing great concern for rules and ecology, a concern we will discuss below. They also emphasized agency and consent on the part of the birds: 'These feathers were found in nature gifted to me by the bird spirits. The talons came to me from a found carcass of the barred owl. Also a gift from the great spirit' (R 21). The same respondent also emphasized mutual exchange, adding that they leave the birds offerings in return. One respondent pointed out that ownership might be a poor way to conceptualize relationships with these objects: 'Own? I humbly carry wings, tails and feathers and feet' (R 8). These are examples of birds acting on humans, building a relationship of gift-giving (see also the Buryat story mentioned in the introduction of this article). They echo scholarly claims that, in many traditional shamanic cultures, there is a responsibility to take only that which is needed or that which is given (Scott 2006). Gift-giving, in this way, is said to confirm the hunter's logic of the relationship with nature as an exchange of vital force between humans and their game (Hamayon 2001). In that dynamic, both parties have agency.

Moreover, during all transformative encounters, an exchange seems to take place between the two entities, in this case the human and the bird, the essence of which ought to be kept secret: one respondent says, 'he does not want me to share' (R 38), another that 'I don't want to share' (R 33) and another that his stories are 'not ones I would share with strangers' (R 32). This secretive exchange is either (1) an exchange of essence or a transformation into the bird of prey during the shamanic journey (R 48); (2) an exchange of mutual recognition which leads to 'clues' (R 37); (3) an exchange of direct communication between species (R 20), possibly reinforcing a special relationship between bird and human (R 15); or (4) an 'unexpected and profound exchange of energy' (R 14). The encounters between birds of prey and respondents are thus not only sensory, but also meaningful and (perhaps mutually) transforming. Also, it should be noted that these transfers primarily take place between an individual human and a bird of prey that is considered a generic representative of its species, though they also involve, to some extent, spirits. Here, in terms of agency, humans seem to establish relationships with birds of prey through their material artifacts or representations. However, the consent for these relationships is given by 'real' birds or spirits, according to our respondents, and the initiative similarly seems to come from the non-human side.

## 5. Religious Ecologies and Relationships of Care

So far, we have discussed birds of prey as embodied symbols and sources of power through several mechanisms of transfer. As some of the examples have shown, mutuality is emphasized as a way of contextualizing and justifying different types of encounters between shamans and birds of prey. In light of this, it is important to consider how relationships between shamans and the ecology around them have been described by scholars. Shamans are said to cultivate intense, intimate and transforming relationships with local lands, animals and lifeforms. These relationships are attributed to the very heart of shamanism, the term itself suggesting shared patterns of expression evident in the relationships practitioners have with self, society and environment (Grim in Walter

and Fridman 2004, p. 107). It is these relationships that we refer to when discussing religious ecology.

These ecologies, however distinctly shamanic, are always in dialogue with the cultural context of the practitioner, whether explicitly or implicitly. We can see this clearly in Core Shamanism, which is embedded in a western context; power animals here usually take the form of dolphins, wolves, eagles or horses, indicating a western perspective on benevolent and/or powerful beings (Harvey and Wallis 2007). The western shamans who answered our questions similarly express an individually selected and contextually dependent combination of views.

In traditional contexts, the mutually reflexive and creative flow between shamans, their communities and other beings is said to give rise to an animistic epistemology. It also implies a religious ecology that stresses the interrelationship of community, local environment and the larger cosmos (Grim in Walter and Fridman 2004). We can recognize this animistic epistemology in how our respondents conceived of the relationship between shamans, birds of prey and spirits. They often described a protective relationship that can go both ways: the bird provides spiritual protection (R 31, 28) and the human in turn provides physical protection. One respondent wrote: 'I care to protect them. I also care about managing pests without poison to protect our birds of prey from accidentally ingesting it' (R 39). Another mentioned being involved in 'campaigns that protect them from the gamekeepers who kill them, or the commercial grouse moors' (R 34). As mentioned before, another described working with non-releasable birds, caring for them while simultaneously being impressed by their power and intelligence (R 36). Perhaps this urge to protect vulnerable species is part of a larger eschatological, apocalyptic or millennial theme. In contemporary shamanic spirituality, Townsend (2004) identified this theme as relating to climate change and the rapid destruction of natural landscapes. This also relates to what Ingold notes as the distinctive feature of animist ontology: the recognition that life is not *in* things but rather that things are *in* life (Harvey and Wallis 2007, p. 31).

Based on our survey responses, this relationship is mutually fostered, although to different ends. Relationships require participants, and in a modern shamanic cosmology, it seems that 'nature' and its different forms, including birds of prey, are considered worthy though not uncomplicated players. According to Townsend, preventing the world's destruction by rekindling lost spiritual awareness and learning from indigenous people is embedded in the modern shamanic spiritual mission (Townsend 2004). Among our respondents, this manifested in the multiple forms of cultural traditions that inspired them. Therefore, these relationships exist across species but also across cultures and even perhaps generations.

These ways of relating propose complex and at times inverted power dynamics between species: it remains striking that, in the stories of our respondents, peaceful cohabitation and mutual aid replaces the traditionally described shamanic imaginary of natural habitat which is not always friendly towards humans. Animals and spirits serve, help and enhance humans, and shamans individually benefit from these relationships. This echoes Gergen's articulation of agency as a relational process: 'to think independently of any relationship is not to think at all' (Gergen 2009, p. 366). In fact, as Gergen states, 'the very idea of individual persons is a byproduct of relational processes' (idem: xxvi). It should be noted, though, that individuality remains important in this dynamic: the contemporary western shaman acts first and foremost on their own behalf, healing, understanding and progressing on an individual path that stands in contrast to more traditional, communal and socially important forms of shamanism (Gergen 2009). Birds offer 'meaning, messages and medicine' (R 49). But unlike in traditional forms of shamanism, these are not necessarily for the community of humanity, for which the shaman acts as an intermediary, but rather for one's own personal benefit. A human community, albeit in many cases an imaginary one, seems important modern shamans, as our respondent were all active on internet fora about shamanism. To take a step further, we might say that contemporary shamans recognize themselves as shamans also, but definitely not exclusively, through

the confirmation their relationships with other species, among which birds of prey, brings about. Relating to birds of prey is a power exchange facilitated by attributes, signification and self-identification as a (modern) shaman.

In neo-shamanism, ecological protection is often framed as a maintenance or restoration of balance. As each entity has its own place in the balance of the world they are all equal in importance. The concept of balance, equivalence of value and lack of hierarchy are linked, for example, in the refusal of respondents to choose a favorite bird of prey. They repeatedly emphasize that 'they each hold their own place, for me' (R 4) and that "all creatures (both predator and prey) have equivalence, there are no 'kings and queens'" (R 34). Another respondent wrote, 'all birds and animals have important medicine and roles in life not so sure there is an emphasis on birds of prey over others [sic]' (R 40). One respondent even shared a story conveying this interdependence and ecological equality between species:

> The Eagle was thought to be the mightiest of all the birds. She could fly higher, see farther and take the largest prey. Hummingbird sought to challenge her with a test of flight. The day of the challenge hummingbird was no where to be found. Still Eagle took to the sky to prove, once again that she was the highest flyer. When she was at her highest altitude she turned to see hummingbird flying along side of her. Amazed, she asked hummingbird how it achieved such a great height. Hummingbird replied that it had ride upon the eagles tail. In this way Eagle saw the interdependence of all living things' (R 8).

It is unclear where this story comes from, but its ethos of balance and interdependence was echoed by eight of our other respondents, and it also fits particularly well in a contemporary shamanic context. Whereas traditional shamanism is usually described as dualistic and conflict-based, relying on the shaman to restore balance, contemporary shamanism often assumes that everything is already, naturally in order and that everything has its place. Indeed, neo-shamanism has much in common with new age spirituality and its pantheistic approach to reality (Townsend 2004). Furthermore, modern shamanism intervenes foremost of the self and is less concerned with intervening into the world. The universe is often conceptualized as benevolent, without a dichotomy between good and evil (Townsend in Walter and Fridman 2004). This helps explain the overwhelmingly negative responses to the question of 'favorites,' as well as the lesson that we ought to respect all animals that wander the earth. In this sense, the relationship to birds of prey clearly reflects the religious ecology that defines modern shamanic spirituality. Although birds of prey are high in the food chain, their relationship with humans in modern times is rarely that of hunter and hunted. This implies that the give and take between humans and these birds is symmetric out of necessity: the relationship is pragmatic and only makes sense if the partnership is sought. We could say that contemporary shamans 'reinvent' (improvise) their relationship with non-humans so as to get in touch with something 'beyond,' be that a 'spirit,' a different version of themselves or another realm. However, as we have highlighted in this section, the humans in this survey are not 'takers' but rather 'receivers' and 'givers'. Besides maintaining and restoring harmony by relating beyond species, the contemporary shaman also finds it important to 'work on the self.'

This brings us to a tension seen throughout the responses between zoological and archetypal interpretations of birds of prey. On the one hand, there is the material view, in which these birds are primarily physical beings limited to and by their physicality. On the other hand, there is the view that they act as symbols of something 'beyond.' In this second interpretation, birds of prey are spirit(s), or are at least related to them or acting as their messengers. In light of these two dimensions, agency must also be explored, since the 'beyond' perhaps implies a more complex relationship of mutuality and exchange. This suggests a unique element of western-based contemporary shamanism, in which agents are simultaneously part of both a materialistic western culture–as real birds, as objects representing birds and as symbols for qualities and, culture that values rational knowledge and physicality–and an animistic shamanic worldview, where birds

are connected to spirit(s). The tension between zoological and archetypal can also be seen as a strain between the socially shared reality and the other realm that is the providence of shamanic spirituality. Respondents themselves considered that other realm flexible and dependent on contextual factors. What a bird of prey 'means' or 'brings' depends on the moment, the specific situation and the personal circumstances of the shaman. And they make sense of this in a personal, individual way. That brings us to a third tension, between personal/individual and communal/social forms of belief and practice. It is striking but not surprising that the responses pointed toward individualized practices, which means our third tension overlaps with the tension between what is identified as modern (Townsend 2004) and more traditional forms of shamanism (Eliade [1964] 1972). Although the religious ecologies of these practices suggest holism and interrelationship, we first of all deal with ways of relating with animal and spiritual others through the individual self. This resonates with Vitebsky's claim that 'cosmology involves not only a vision of how the universe works, but also uses this as a basis for decisive action upon the world' (Vitebsky 1995, p. 199). Meanwhile, holism 'becomes just one value, rather than the ground for all other values—and so, becomes no more than an option' (Vitebsky 1995, p. 205).

This finding aligns with a core principle of animist ontologies underlying shamanism—that these ontologies are primarily relational, based on the human need to communicate, as the authors of a Special Issue of *Ethnos* agree (Ingold 2006; Hornborg 2006; Bird-David 2006; Scott 2006). However, in order to communicate, a relationship of equality is necessary, one in which both sides are active subjects (Rival 2012). The act of communication, real or perceived, thereby elevates non-humans from objects to subjects (Rival 2012), as we can see in many examples from our respondents. This warrants a revision of Callon and Law's definition of agency cited above, which places language-use front and center. Instead, we propose that communication is a primary condition for the experience of agency, whether or not this communication uses (human) language.

We should also heed Hornborg's call to pay attention to ontologies, which differ from the fundamental assumptions of Cartesian science. Nonetheless, 'engagement' and 'relatedness' exist alongside 'detached observation,' and for Hornborg, these categories delimit animist thought and are context-dependent. This suggests that our subjects navigate between ontologies and are "struggling for 'relatedness'—for a restoration of meaning." At the same time, they remain confined to the 'modernist . . . vocabulary through which objective proprieties are attributed to distinct, external things' (Hornborg 2006, p. 26).

### 6. Conclusions

This article has attempted to map the relationships between contemporary shamans and birds of prey, identifying how different practitioners understand the importance and meaning of these animals and the experiences they have together. Transfer between species, either human and animal or the more complicated triad of human–animal–spirit, has been the main finding of this inquiry. In our survey responses, we see a transfer of identity and/or power (or an exchange of vital energy, to use Ingold's words (Ingold 2000)). In terms of the transfer of power, we are able, through different forms of encounter, to identify the agency of human, bird and spirit, which differs per situation and per respondent. In terms of transfer of identity, we see a spectrum of different modalities of relating, starting with the observation of desirable characteristics in non-humans, moving to the embodiment of those characteristics, and arriving at the human (temporarily) 'becoming' the bird of prey.

Situating birds of prey within contemporary shamanism is particularly difficult because it simultaneously concerns real-life birds as well as the metaphors or archetypes the shamans engage with. To the researcher, it is not always clear which of these categories is relevant for a given respondent. In fact, perhaps the same goes for the respondents themselves. More importantly, this tension is an invitation to reconsider the epistemological separation between these categories. The physical animals seem to bear their metaphorical

meaning as well as the possibility of somehow transferring its associated archetypal abilities to the shaman.

We have found that transformative interactions between shamans and birds (of prey) in contemporary western-based shamanism are best understood as instances of transfer. Based on Ojamaa's model, we recognize objective and expressive transformations in our respondents' descriptions, but we also note that most instances described move beyond an anthropocentric dialectic and toward a more plural and multivocal model of signification. Here, the shaman has agency in their interpretation and reception of messages. The birds also have agency as catalysts of the shaman's transformation (of consciousness) and in their attunement to the spirit world.

The modalities of transfer we have identified are *transfer through emulation*, *transfer through encounter* and *transfer through physical touch*. These modalities each involve different degrees of physical and spiritual closeness between human and animal, but also implicitly between human, animal and spirit. In these, it matters tremendously which aspects or characteristics of the birds of prey are considered important. It also matters how they view their own activities and whether they are inspired by what they view as traditional shamanism, and both these factors partially inform the 'use' or content of transfer. These modalities align with Harvey's ideas on animist relationality and copresence (Hornborg 2006, p. 15). We would also like to stress again the individuality of our respondents' shamanic visions and practices. Although most subscribed to a holistic religious ecology, where humans and animals like birds of prey are part of a complex interrelationship, intervention and action within this ecology happens mostly through work on the self. Work on the self is closely connected to the western understanding of agency as individual self-realization and autonomy (Mahmood 2005). Furthermore, most respondents believed they were operating in a benevolent, harmonious universe in which action is not crucial (for survival), but where action is nonetheless welcomed. It is worth pointing out that this harmonious, benevolent worldview is common in New Age spirituality, but less so in shamanism. In fact, traditional shamanism is often described as being survival-oriented and battling the dangerous forces of nature. Our respondents do not seem keen to adhere to that description. This shows that contemporary western-based shamanism is a varied and unique amalgamation of traditional and modern practices, and that no homogenous 'shamanism' exists.

**Author Contributions:** Conceptualization, C.I. and N.G.; Theoretical background, C.I. and N.G.; Methodology, C.I.; Data collection, C.I.; Data curation, C.I. and N.G.; Formal analysis, C.I. and N.G.; Writing–original draft, C.I. and N.G.; Writing–review and editing, C.I. and N.G. All authors have read and agreed to the published version of the manuscript.

**Funding:** This research received no external funding.

**Informed Consent Statement:** Informed consent was obtained from all subjects involved in the study.

**Data Availability Statement:** The data used for this article is available upon request.

**Conflicts of Interest:** The authors declare no conflict of interest.

## Appendix A. Survey Questions

1. Are you: (shaman, shamanic practitioner/other (if possible, please explain)
2. Did you receive a formal education as a shaman? y/n/other
3. What comes to mind with 'birds of prey'?:
4. Which birds do you associate with 'birds of prey'?:
5. In the tradition of shamanism that you are affiliated with, is there any importance given to (specific) birds of prey?:
6. Would you be able to share a bit about the shamanic tradition you belong to?

7.  Do you have a favorite bird of prey?
8.  Do you have a personal story relating to this bird?
9.  Are there any stories that you know of connected to birds of prey that you could share with us?:
10. Are birds of prey personally important to you?
11. In which way, if any, are birds of prey present in your everyday life?
12. Do you own any bird-related items?
13. Do you consider seeing birds of prey a sign? If so, how do you figure out what it means?
14. What is your age?

    ○  Under 12 years old
    ○  12–17 years old
    ○  18–24 years old
    ○  25–34 years old
    ○  35–44 years old
    ○  45–54 years old
    ○  55–64 years old
    ○  65–74 years old
    ○  75 years or older

15. What is your gender?

    ○  Male
    ○  Female
    ○  Other, namely

16. What is the highest degree or level of school you have completed? If currently enrolled, highest degree received.

    ○  No schooling completed
    ○  Nursery school to 8th grade
    ○  Some high school, no diploma
    ○  High school graduate, diploma or the equivalent (for example: GED)
    ○  Some college credit, no degree
    ○  Trade/technical/vocational training
    ○  Associate degree
    ○  Bachelor's degree
    ○  Master's degree
    ○  Professional degree
    ○  Doctorate degree

17. Do you live in . . .

    ○  in a city
    ○  in a smaller town
    ○  in the country side
    ○  elsewhere, namely

18. Which country do you live in?
19. Is the shamanic tradition you practice native to the country you live in/
20. Is there anything else that you would like to add?
21. Would you like to participate in follow-up research on the same topic? Would you like to receive more information about the results of this research? If yes, please let me know on which email address I can reach you.

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
