# Peer review of "Birds of Prey, Birds of Wisdom: Relating to Non-Humans in Contemporary Western-Based Shamanism"

_religions, doi:10.3390/rel13121214_

Round 1

Reviewer 1 Report

BROAD COMMENTS

This is a potentially interesting article that, once revised, could contribute to the literature on contemporary shamanism in Western countries, by widening the knowledge of practices that are increasingly popular. However, to reach this, I believe that a major revision is needed.

First, I would suggest the Authors to better contextualize their topic, by giving more space to the critical analysis of neo-shamanism and placing the topic in the context of the so-called ‘new spirituality’. Connecting to this, Townsend’s definition of ‘modern shamanic spirituality’ should be clearly explained.

It is also necessary to better contextualize and problematize Eliade’s discourse, as his definition of ‘shaman’ is essentialistic and thus needs to be presented in a critical way. The Authors could shift the focus from Eliade’s definition of ‘shaman’ to the influence that his work had on the so-called neo-shamanisms.

Moreover, the category of shamanism needs to be problematized. Study shows that the concept of ‘shamanism’ is a construct (some would say an ‘invention’) that appeared at the end of the 19th century. Van Gennep already pointed this out in 1903. See also Ioan Lewis (2003) and Znamesky (2007).

Three considerations on the use of the sources:

-        Pratt 2007 is not an academic source and, thus, should not be used: the author is a practitioner, not a scholar.

-        Walter and Fridman 2004 is an edited volume. As such, in-text references need to present the author of the specific contribution in volume the Authors are referring to.

-        Page numbers should be included.

In the section ‘References’ the Authors do not always follow the guidelines, especially for what concerns the indication of the authors’ names/surnames.

SPECIFIC COMMENTS

· The title presents the concept of “Western-based Shamanism”. In their discussion, the Authors should clarify which Western countries they are referring to.

·     Lines 7-8 and lines 87-88: Dolphins, wolves and horses are not “birds of prey” and cannot be presented as an example of the fact that “It is known that birds of prey have their place in contemporary forms of shamanism”.

·     Line 11 and line 91: “in which way” and “how” are synonyms.

·     Line 58: The Authors should provide at least an example of which “shamanic things” they refer to are “Western in origin”.

·     Line 64: The Authors write that they will “use this term, modern shamanic spirituality, throughout this article”. This has been used only once.

·     Line 71: The meaning is not clear.

·     Lines 74-75: The Authors write they will keep “a close eye to the way agency is expressed, attributed, or explained”. I would suggest placing more emphasis on this aspect in the course of the article.  

·     Lines 78-79: If the Authors take the use of language as one condition of agency, then their birds of prey would show no agency.

·     Line 102: I would suggest offering a more accurate picture of these respondents: where are they from? What is their age? Gender? Level of education?

·     Line 102: I would suggest clarifying the difference between “shamans” and “shamanic practitioners”. In the following pages they seem to be used interchangeably.

·     Lines 107-108: This sentence needs to be rephrased for clarity.

·     Line 159: How many respondents are part of this “largest group”? Have the Authors considered that their answers might depend on Harner’s teachings?

·     Line 162: How many are these “many” who mix different traditions?

·     Line 164: How many are part of this “smaller group?

·     Line 175: In what sense the balancing is “delicate”?

·     Line 177: How many are these “some respondents”?

·     Line 217: To which tradition do the “traditional shamans” the Authors refer to belong? This should be clarified to avoid excessive generalisations.

·     Lines 231-234: Considering that most of the respondents are connected to Harner’s Core Shamanism, a shamanism based on South American shamanic practices, why did the Autors chose to use the model Ojamaa built to study Siberian shamanism? This should be explained.

·     Line 247: “note the agency”: whose?

·     Line 256: Which “forms of traditional shamanism” do the Authors refer to? This should be clarified to avoid excessive generalisations.

·     Line 314: How do birds of prey cultivate these relations?

·     Line 328: How many respondents are “most respondents”?

·     Line 329: Which “traditional shamanism” do the Authors refer to? This should be clarified to avoid excessive generalisations.

·     Lines 386-390: I would suggest that the Authors rephrase to better clarify the three types of exchange they are talking about.

·     Line 396: With whom are these “relationships” established?

·     Lines 407-411: This is an excessive generalization.

·  Line 417: “The western shamanic practitioners”: were there non-Western respondents to the survey? This should be clarified.

·     Lines 418-419: I would suggest expanding on this.

·     Line 422: “a relational knowing as if between distinct persons”: I suggest that the Authors rephrase this for clarity.

·     Line 434: Townsend 1984 does not appear in the reference list.

·     Line 439-443: This part should be reworded to clarify its meaning.

·  Line 450: What do the Authors refer to when they speak of “traditional shamanism’s imaginary of salvage and survival-oriented nature”?

·     Line 463: I would suggest that the Authors consider that shamans also need a (albeit 'imaginary') community that recognises them as shamans.

·     Line 467: “In present-day shamanic practices”: traditional or ‘new’?

·     Line 468: “restoring balance and harmony”: the examples presented below do not support this claim.

·     Lines 482-483: “its message of harmony…by many of our respondents”: does this mean that many of your respondents (how many?) made reference to this story?

· Lines 487-488: “much in common with Neopaganism and New Age spirituality”: this point should be expanded to make the paper more engaging.

·  Line 495: “seems to contradict Hamayon’s explanation”: please consider that Hamayon refers to another type of shamanism.

·     Line 506: “for goals beyond this realm”: what does this expression mean?

·  Line 518: In what sense are birds of prey “part of a materialistic western culture”?

·    Line 561: Throughout the paper, the topic of “spirits” appears only marginally and should be expanded upon.

Author Response

Dear reviewer,

Thank you for your detailed and attentive review, it has been much appreciated. All comments have been extremely helpful in revising this paper and we would like to respond to your suggestions in detail, as we much appreciated also the manner of reviewing.

We agreed especially that we needed to better contextualize and problematize the topic of shamanism to avoid excessive generalizations, to be more precise about different ethnic groups and their relationship to birds of prey and better explain the difference between categories used, such as traditional shamanism, neo-shamanism and modern shamanism, just to insist on the most important ones. We both agree that there is no unified category of ‘traditional shamanism’ and use this throughout the paper an emic concept, but upon re-reading we see that this did not come through as intended. Hence, we have added a few paragraphs to the introduction, in which we explain that ‘traditional shamanism’ is a useful term for us insofar it informs contemporary (neoshamanic) practice, but that as a scholarly concept, it is not  reliable. Throughout the article, we now say ‘traditional shamanism is described/imagined’ rather than ‘in traditional shamanism, …’. In the introduction and throughout, we have also placed neoshamanism more firmly in a new age context, and we have expanded on Townsend’s ‘modern shamanic spirituality’.

Three considerations on the use of the sources and references: we have addressed these issues, hopefully nothing escaped our attention. 

SPECIFIC COMMENTS

  • The title presents the concept of “Western-based Shamanism”. In their discussion, the Authors should clarify which Western countries they are referring to.

We have now included more demographic data about our respondendts, including country of residence and, to a degree, ethnic belonging.

  • Lines 7-8 and lines 87-88: Dolphins, wolves and horses are not “birds of prey” and cannot be presented as an example of the fact that “It is known that birds of prey have their place in contemporary forms of shamanism”.
  • Line 11 and line 91: “in which way” and “how” are synonyms.
  • Line 58: The Authors should provide at least an example of which “shamanic things” they refer to are “Western in origin”.
  • Line 64: The Authors write that they will “use this term, modern shamanic spirituality, throughout this article”. This has been used only once. 
  • Line 71: The meaning is not clear. 

Thank you, all these problems are now addressed.

  • Lines 74-75: The Authors write they will keep “a close eye to the way agency is expressed, attributed, or explained”. I would suggest placing more emphasis on this aspect in the course of the article.  

This is one of the very few comments with which we slightly disagree. We think that agency does come back in the course of the article, however it may not always have been clearly formulated. The article has now, with the help of a colleague, undergone major language revisions, and we hope that this line is now more clearly in the foreground throughout this article.

  • Lines 78-79: If the Authors take the use of language as one condition of agency, then their birds of prey would show no agency. 

We have made this tension explicit and suggested communication as an alternative for language.

  • Line 102: I would suggest offering a more accurate picture of these respondents: where are they from? What is their age? Gender? Level of education?

As mentioned above, we introduced our respondents now with the demographic data we have on them, and we believe this paints a clearer picture of them, so thank you for this suggestion.

  • Line 102: I would suggest clarifying the difference between “shamans” and “shamanic practitioners”. In the following pages they seem to be used interchangeably. 

This suggestion provided an opportunity to address the self-identification of our respondents, which varied between these two terms and others. We chose to use ‘shaman’ throughout the article for clarity and have explained this choice in the text.

  • Lines 107-108: This sentence needs to be rephrased for clarity. 
  • Line 159: How many respondents are part of this “largest group”? Have the Authors considered that their answers might depend on Harner’s teachings? 
  • Line 162: How many are these “many” who mix different traditions? 
  • Line 164: How many are part of this “smaller group?
  • Line 175: In what sense the balancing is “delicate”?
  • Line 177: How many are these “some respondents”?

Thank you, fixed.

  • Line 217: To which tradition do the “traditional shamans” the Authors refer to belong? This should be clarified to avoid excessive generalisations.
  • Lines 231-234: Considering that most of the respondents are connected to Harner’s Core Shamanism, a shamanism based on South American shamanic practices, why did the Autors chose to use the model Ojamaa built to study Siberian shamanism? This should be explained. 
  • Line 247: “note the agency”: whose? 
  • Line 256: Which “forms of traditional shamanism” do the Authors refer to? This should be clarified to avoid excessive generalisations.

These questions tie in with the broad comments, where we have formulated a response. We hope that there are no more excessive generalisations in the article.

  • Line 314: How do birds of prey cultivate these relations? 

Addressed

  • Line 328: How many respondents are “most respondents”?

All these suggestions have been met by including the number in parantheses.

  • Line 329: Which “traditional shamanism” do the Authors refer to? This should be clarified to avoid excessive generalisations.

Fixed, see comments above.

  • Lines 386-390: I would suggest that the Authors rephrase to better clarify the three types of exchange they are talking about. 

Thank you, fixed.

  • Line 396: With whom are these “relationships” established?

We made this more clear now.

  • Lines 407-411: This is an excessive generalization.

Thank you, see comments above.

  • Line 417: “The western shamanic practitioners”: were there non-Western respondents to the survey? This should be clarified.

Fixed, see above.

  • Lines 418-419: I would suggest expanding on this.
  • Line 422: “a relational knowing as if between distinct persons”: I suggest that the Authors rephrase this for clarity. 
  • Line 434: Townsend 1984 does not appear in the reference list.
  • Line 439-443: This part should be reworded to clarify its meaning.
  • Line 450: What do the Authors refer to when they speak of “traditional shamanism’s imaginary of salvage and survival-oriented nature”?

We found all these comments helpful and addressed them in the article.

  • Line 463: I would suggest that the Authors consider that shamans also need a (albeit 'imaginary') community that recognises them as shamans.

Yes, thank you for the reminder

  • Line 467: “In present-day shamanic practices”: traditional or ‘new’?
  • Line 468: “restoring balance and harmony”: the examples presented below do not support this claim.
  • Lines 482-483: “its message of harmony…by many of our respondents”: does this mean that many of your respondents (how many?) made reference to this story?

Thank you, the comments above have been addressed.

  • Lines 487-488: “much in common with Neopaganism and New Age spirituality”: this point should be expanded to make the paper more engaging.

We did, see response to broad comments.

  • Line 495: “seems to contradict Hamayon’s explanation”: please consider that Hamayon refers to another type of shamanism.

Indeed, we have checked and removed the lines referring to this. Mea culpa

  • Line 506: “for goals beyond this realm”: what does this expression mean?
  • Line 518: In what sense are birds of prey “part of a materialistic western culture”?

 Thank you, we clarified both.

  • Line 561: Throughout the paper, the topic of “spirits” appears only marginally and should be expanded upon.

This is one of the very few comments with which we slightly disagree. As with agency, we think spirit(s) does/do come up throughout the article, however it may not always have been clearly formulated. After the major language revisions, we hope that this line is now more clearly in the foreground throughout this article.

Reviewer 2 Report

This is a well-researched and absorbing paper. 

I draw attention to the following points that merit attention. 

Par. 4 on 5, beginning at line 212 builds up to a reference to Turner 1991: 147. This item does not, however, appear in the References. I assume this is Victor Turner on ritual and performance, and that the term 'arara' in the quotation may refer to Arará Afro-Cuban people originating in Dahomey. This citation is important because it is used to underpin the major and not wholly uncontroversial claim that 'the semiotic distinction between signified and signifier becomes obsolete for the shamanic practitioner' (lines 213-4).   

Line 254 correct 'eminding me...' 'to 'reminding me...'. 

Line 480 what tense is 'ride', should this not be 'rode' or 'had ridden...'?

Lines 484 ff. The bare assertion of a summary juxtaposition of 'traditional' and 'modern' shamanism as 'clearly dualistic' versus 'monistic', respectively, may well attract a critical response. 

Line 586, insert word: 'most respondents believe themselves to be operating...'. 

Author Response

Dear reviewer,

Thank you for your attention to this article and your kind words. We have addressed your comments in the following ways.

  • Turner (1991) is referring to Terrence S. Turner, and the reference was already in the list.
  • Language has been revised and cleared up, the typos are no longer in the article - thank you for pointing them out!
  • Based on another reviewer’s comments, we have significantly changed our approach to traditional shamanism, so broad generalizations such as these are hopefully no longer in the article.

Round 2

Reviewer 1 Report

The Authors have addressed all the issues raised, greatly improving their article.